# Epidural Stimulation and Resistance Training (REST-SCI) for Overground Locomotion After Spinal Cord Injury: Randomized Clinical Trial Protocol

**DOI:** 10.3390/jcm14061829

**Published:** 2025-03-08

**Authors:** Ashraf S. Gorgey, Robert Trainer, Refka E. Khalil, Jakob Deitrich, Muhammad Uzair Rehman, Lance L. Goetz, Denise Lester, Adam Klausner, Carrie L. Peterson, Timothy Lavis

**Affiliations:** 1Spinal Cord Injury and Disorders, Richmond VA Medical Center, Richmond, VA 23249, USA; refka.khalil@va.gov (R.E.K.); deitrichjn@vcu.edu (J.D.); rehmanu@vcu.edu (M.U.R.); lance.goetz@va.gov (L.L.G.); timothy.lavis@va.gov (T.L.); 2Department of Physical Medicine & Rehabilitation, Virginia Commonwealth University, Richmond, VA 23298, USA; robert.trainer@va.gov (R.T.); denise.lester@va.gov (D.L.); adam.klausner@vcuhealth.org (A.K.); 3Department of Anesthesiology, Richmond VA Medical Center, Richmond, VA 23249, USA; 4Department of Biomedical Engineering, Virginia Commonwealth University, Richmond, VA 23284, USA; clpeterson@vcu.edu; 5Department of Urology, Richmond VA Medical Center, Richmond, VA 23249, USA

**Keywords:** epidural stimulation, spinal cord injury, exoskeleton, standing, locomotion, neuromuscular electrical stimulation, body composition, overground ambulation

## Abstract

**Introduction:** Implanted spinal cord epidural stimulation (SCES) is an emerging neuromodulation approach that increases the excitability of the central pattern generator [CPG] and enhances tonic and rhythmic motor patterns after spinal cord injury (SCI). We determine the effects of exoskeleton-assisted walking [EAW] + epidural stimulation [ES] + resistance training [RT] on volitional motor control as a primary outcome, as well as autonomic cardiovascular profile, body composition, and bladder function compared to EAW + delayed ES + noRT in persons with motor-complete SCI AIS A and B. **Methods and Analysis:** Twenty male and female participants [age 18–60 years] with traumatic motor-complete SCI [2 years or more post injury], and level of injury below C5 were randomized into either EAW + ES + RT or EAW + delayed-ES + no-RT groups for more than 12 months. Baseline, post-interventions 1 and 2 were conducted six months apart. Measurements included body composition assessment using anthropometry, dual x-ray absorptiometry, and magnetic resonance imaging prior to implantation to evaluate the extent of spinal cord damage, neurophysiologic assessments to record H-reflexes, overground ambulation and peak torque for both groups, and the Walking Index for Spinal Cord Injury Scale [WISCI 2]. Metabolic profile measurements included the resting metabolic rate, fasting biomarkers of HbA1c, lipid panels, total testosterone CRP, IL-6, TNF-α, plasma IGF-I, IGFBP-3, and then a glucose tolerance test. Finally, urodynamic testing was conducted to assess functional bladder improvement due to ES. **Results:** The restoration of locomotion with ES and EAW may result in a reduction in psychosocial, cardiovascular, and metabolic bladder parameters and socioeconomic burden. The addition of the resistance training paradigm may further augment the outcomes of ES on motor function in persons with SCI. **Conclusions:** Percutaneous SCES appears to be a feasible and safe rehabilitation approach for the restoration of motor function in persons with SCI. The procedure may be successfully implemented with other task-specific training similar to EAW and resistance training.

## 1. Background

Spinal cord injury (SCI) is a devastating medical condition that involves the disruption of motor, sensory, and autonomic functions. In the U.S., the prevalence of survivors with SCI is 256,000–400,000, with an estimated 11,000–12,000 new cases annually and an 18% growth in prevalence since 1988 [1,2]. Furthermore, there are an estimated 46,000–65,000 veterans with SCI. The social and economic burdens are increased by escalating charges for each patient that range from USD 342,000 to USD 1,048,000 for stabilization and rehabilitation after SCI, with annual charges ranging from USD 41,000 to USD 182,000 [2,3,4]. The estimated lifetime costs of SCI are USD 7.3 to 12 million per person [2,3,4]. Additionally, advances in the healthcare system have resulted in the increased longevity of persons with SCI, but with a longer lifespan associated with several costly chronic comorbidities as well as poor quality of life [5,6,7].

For years, several research groups have focused on the restoration of locomotion to ameliorate comorbidities and secure independence after SCI [8,9,10,11,12,13,14,15,16,17]. Improved mobility would result in a reduction in psychosocial, cardiovascular, and metabolic parameters and also in socioeconomic burden [5,6,7]. Prior attempts with locomotor training (LT) have included the use of long leg braces, hip–knee–ankle–foot orthoses, functional electrical stimulation (e.g., Parastep^®^ (Sigmedics Inc., Fairborn, OH, USA)), and therapist-driven or robotic-driven body weight-supported treadmill training (BWSTT) [8,9,10,11,12,13,14,15,16,17]. Previously, LT using BWSTT has also been shown to improve multiple bladder parameters [18]. Unfortunately, none of these approaches has been successful in restoring overground mobility because they are metabolically demanding and result in premature fatigue during walking in persons with SCI [8,9,10,11,12,13,14,15,16,17]. Another major concern is that the current paradigm of restoring locomotion does not consider the dramatic effects of muscle atrophy after SCI. The skeletal muscle cross-sectional area (CSA) can be as low as 50% compared to able-bodied (AB) controls [18]. Spungen reported that monozygotic twins with acquired paraplegia had lower fat-free mass than their AB twins [19]. Resistance training (RT) has been identified as a very important rehabilitation approach that increases lean mass and decreases fat mass [20,21]. In persons with chronic SCI, surface neuromuscular electrical stimulation (NMES-RT) has been recommended as an effective rehabilitation strategy to restore muscle size [22,23].

Implanted spinal cord epidural stimulation (SCES) is an emerging neuromodulation approach that may increase the excitability of the central pattern generator (CPG) and enhance tonic and rhythmic motor patterns in persons after SCI [22,23,24,25,26] and result in the activation of lower extremity muscles for standing and stepping with and without assistive devices. Several case reports have successfully demonstrated that ES results in evoking motor potentials in the paralyzed muscles of persons with motor-complete and motor-incomplete SCI (C5-T8; AIS A, B, and C). Evoked motor potentials occurred in both supine and standing positions using stimulation intensities ranging from 0.5 to 10 V, frequencies of 2–30 Hz, and pulse durations of 210–450 µs to recruit both proximal and distal leg muscles (Table 1). Multimodal neuromodulation intervention in conjunction with task-specific training provided promising outcomes in the restoration of motor behavior and overground locomotion in persons with SCI [22,23,24,25,26,27,28,29,30,31,32]. Manual-assisted body weight-supported treadmill training (BWSTT) is frequently used as task-specific training accompanied with overground ambulation to improve stepping and walking after SCI [9,10,11,12,13]. Despite the absence of supraspinal control, manual BWSTT enhances sensorimotor recovery and modulating spinal excitability via sensory and kinesthetic feedback to the cord through the muscles, tendons, and joints of the lower extremities [8,9,10,11,12,13,22,23,24,25]. However, BWSTT is considered a labor-intensive rehabilitation approach that requires at least three to four specialized or trained therapist to ensure the successful completion of a single session with one patient [8,9,10,11,12,13]. This limits the wide application of this task-specific training in the clinical setting. On the contrary, exoskeleton-assisted walking (EAW) has become widely accessible and may need only one trained person to ensure the successful completion of overground locomotor training [33,34,35,36]. Overall, EAW reduces the metabolic cost of walking and provides the opportunity to exercise easily for more than 60 min [37,38]. Additionally, several studies have utilized EAW as a platform in conjunction with neuromodulation modalities to enhance motor recovery in persons with SCI [39,40,41].

Autonomic and cardiovascular issues are well known in persons with SCI at or above the T6 neurologic level [42,43]. Thus, the majority of participants in this trial with injury levels in the range C5-T10 may present with varying degrees of cardiovascular or autonomic impairments [42,44]. Previous work has shown that many individuals with SCI may show signs of orthostatic intolerance despite their level of injury, although they do not meet the clinical definition of hypotension [44,45]. Thus, examining the effect of spinal cord ES on cardiovascular parameters may enhance our knowledge of the safety and efficacy on autonomic dysfunction in this population [46,47,48,49]. Additionally, bladder dysfunction is a common problem that leads to urinary tract infection and frequent hospitalization after SCI [46,50]. The effects of different neuromodulation techniques are currently under investigation to attenuate the consequences of bladder dysfunctions after SCI [47,48,49,51]. Herrity et al. showed that ES improved urodynamic parameters in a 31-year-old SCI person with C5 AIS B via parasympathetic activation [49]. Following urodynamic studies, accompanied with pelvic floor muscle surface EMG, voiding efficiency increased from 21.9% to 68.5%, and after 6 months, the voiding efficiency was 27.2% higher without using ES [49].

Therefore, a complementary program that ensures the loading of the paralyzed skeletal muscles via NMES-RT and maximizes afferent loops to locomotor centers may enhance the outcomes of ES on motor recovery [52,53]. The aim of this study is to maximize functional motor gains by considering a multi-level rehabilitation approach (i.e., ES + RT) compared to a single approach (i.e., ES only) (Figure 1). The objectives of this pilot clinical trial are to examine the effects of EAW + ES + RT on the volitional motor control (primary endpoint variables), autonomic cardiovascular profile, body composition (secondary outcome variables), and bladder function (exploratory variable) compared to EAW + delayed ES + no RT in persons with motor-complete SCI AIS A and B. We propose a simple rehabilitation approach using EAW + ES with RT (EAW + ES + RT) for 12 months to enhance volitional motor recovery and enhance EAW overground stepping ability in persons with motor-complete SCI. This may further decrease the time, number of providers, and cost necessary to enhance overground ambulation, attenuate cardiovascular risk factors, and improve bladder function. Our primary objective is to enhance gait parameters as demonstrated by increasing the 10 m walking speed, unassisted EAW overground walking stepping, and EMG activity compared to just using EAW + ES.

## 2. Methods

### 2.1. Study Design

The study design is shown in Figure 2. Twenty participants are randomly assigned to 13 months of EAW + ES + RT (n = 10) versus EAW+ delayed-ES + no-RT (control group) (n = 10). This includes three measurement weeks (baseline, post-interventions 1 and 2), two weeks for implantation, and twelve months for training. Measurements will be approximately 6 months apart to capture the changes in the outcome variables as a result of training. The EAW + ES + RT group will complete six months of supervised EAW + ES (3× per week) followed by another six months of EAW + ES (3× per week) with progressive RT twice weekly [41,53,54]). In the EAW + ES + RT group, RT will be administered in an open kinematic chain approach for 12 weeks using surface NMES with ankle weights followed by 12 weeks (2× per week) of gradually using the implanted ES to perform the sit-to-stand approach (i.e., using their body weight to load exercising muscles in a closed kinematic fashion) [20,21]. The closed kinematic RT approach facilitates additional activity-based plasticity via increasing afferent firing to the lumbosacral segment [53]. The control EAW + delayed-ES + no-RT group will be enrolled in 6 months of EAW without ES (3× per week), and then this is followed by an additional 6 months (3× per week) of EAW + ES (i.e., delayed entry approach) without conducting RT and performing either passive movement of passive stretching (2× per week) in seated position [55]. This allows for the balancing of the design and accounts for any interaction between both groups. The current design guarantees answers for the two primary research questions of the trial. Both groups would complete two days of assessment using the same sequence outlined in the design at baseline, post-intervention 1, and post-intervention 2. Participants will then undergo a physical activity and quality of life (QOL) questionnaires at the three timepoints of this study [56].

The study design includes 4 main phases for any of the 20 participants for either EAW + ES + RT or EAW + delayed-ES + no-RT. Phase I comprises screening, consenting, measurements, and two weeks of implantation, temporary and permanent implantation. In Phase II, participants in both groups start to exercise with the exoskeleton device three times a week for six months. In Phase III, the EAW + ES + RT group receives electrical stimulation resistance training accompanied with ankle weights twice a week for three months while sitting in their wheelchair. For another three months using the implanted device, participants perform resistance training through a sit-to-stand activity and with a standard walker. The EAW + delayed-ES + no-RT group perform passive movement training twice a week for 3 months. The study design includes 4 main phases for all of the twenty participants for either EAW + ES + RT or EAW + delayed-ES + no-RT.

### 2.2. Screening and Consenting

All twenty participants will undergo motor control studies, cardiovascular and body composition assessments, and urodynamic studies. After informed consent is provided, a physiatrist board-certified in SCI medicine performs a full physical examination, including neurological assessment and AIS examination. Measurements include EAW fitting, 10 m EAW speed, the number of unassisted steps during EAW, EMG pattern from sitting, standing, and walking, H-reflex from sitting, body composition assessment using anthropometry, DXA, metabolic profile, and urodynamic studies.

### 2.3. Randomization and Allocation

Randomization will be conducted following the baseline assessment period using the n-query computer program. Participants will be assigned into either the EAW + ES + RT or EAW + delayed-ES + no-RT group for more than 12 months. Stratified randomization will be considered based on the level of injury and time since injury. The process of randomization will be carried out in advance by our biostatistician and after assigning each participant a sequential ID code (0881, 0882, 0883, etc.). Using a random pattern generator (n = 0881–08820) computer software, the numbers will be evenly randomized into the 2 groups (Table 2).

#### 2.3.1. Implantation Procedures

The implantation of the stimulator is a 2-step process where temporary implantation precedes permanent implantation. The ES system (Intellis Epidural Stimulator, Medtronic, Minneapolis, MN, USA) is used to electrically stimulate lumbosacral enlargement [40,54,57]. During temporary implantation, two 8-electrode lead arrays are implanted utilizing fluoroscopic guidance over spinal segments T10-L2 [40,54,57]. All spinal ES operations and procedures will be performed by an anesthesiologist who will be responsible for implanting and operating the device. Prior to both temporary and permanent procedures, Hibiclens^®^ (chlorhexidine) soap skin cleanser and Bactroban^®^ (mupirocin) 2% ointment are applied for 7 days to reduce bacterial colonization of the participants’ skin and nares. Prior to entering the operating room, consent will be obtained from the participant, and an anesthesia preoperative evaluation will be conducted. In a rare event when opioids are prescribed for more than 5 days, monitoring is undertaken according to clinical recommendations.

MRI

Prior to implantation, participants undergo magnetic resonance imaging (MRI) of the spine to evaluate the extent of spinal cord damage [54,57]. This is a helpful clinical scan to ensure the patency of the spine and appropriate implantation procedures. The MRI scans of the spinal cord are further analyzed to determine the axial damage ratio and lesion length in order to assess the extent of damage as well as for the prognostication of motor recovery in persons with SCI (Figure 3a).

Briefly, a General Electric 3.0 T signa Discovery MR750 Scanner is used to acquire MRI using a two-dimensional fast-spin echo sequence (slice thickness = 5 mm, slice spacing = 10 mm, field of view = 200 × 200 mm^2^, matrix size = 320 × 224, repetition time = 9015 ms, echo time = 104 ms). Fifteen axial images of the spinal cord are acquired using a two-dimensional fast relaxation fast-spin echo sequence (slice thickness = 3 mm, slice spacing = 4.5 mm, field-of-view = 360 × 360 mm^2^, matrix size = 512 × 512, repetition time = 2939 ms, echo time = 95 ms). The spinal cord lesion hyperintensity is segmented throughout each axial slice, followed by the surrounding spinal cord, using OsiriX MD (Pixmeo SARL, Bernex, Switzerland) for the axial damage ratio biomarker. The axial damage ratio is calculated for each slice as Area_lesion_/Area_cord_, and the largest ratio was selected for each participant. The midsagittal T2-weighted image is used to measure lesion length as the most cranial to most caudal ends of the lesion hyperintensity.

b.Temporary Implantation

The implantation process comprises a temporary implantation (to ensure proper lead placement) as well as permanent implantation. It is possible that unforeseen events may emerge after temporary implantation that may lead the patient to withdraw or be considered a screen failure. The patient may also deny participation as result of discomfort or pain during this period of temporary implantation (Figure 3b1). Temporary lead placement may not be possible due to unforeseen epidural space anatomical issues. The temporary leads are cheaper than the permanent leads and provide us with the opportunity to resolve unanticipated events and make the correct medical decision prior to permanent implantation. Two 14-gauge epidural needles are used to access the epidural space using the loss of resistance technique with X-ray assistance. Leads are inserted in the epidural space, and configurations necessary to evoke motor potentials are tested as indicated by visible motor contractions of the paralyzed muscles. The entire procedure is performed in the presence of a Medtronic representative. Confirmation of posterior epidural lead placement is ensured by live lateral and antero-posterior fluoroscopy as the leads would be guided to either side of the midline. Lead position is optimized after proper rudimentary motor stimulation is confirmed during the trial in real time. After lead placement, the needles are removed from the epidural space. Electrodes are to be secured to the skin with tape and glue to limit lead migration. The goal of temporary implantation is to confirm the activation of the lumbosacral segments before proceeding with permanent implantation. After implantation, spinal mapping is conducted for 3–5 consecutive days, beginning on either the same or the following day.

c.Permanent Implantation and Recovery

Seven days after temporary implantation, two 8-electrode arrays of Vectris leads are implanted in an operating room [41,54,57]. Phase 1 of permanent implantation is similar to that of the temporary procedure. but after an anesthesiologist administers anesthesia. An IV line is established, and standard American Society of Anesthesiologists (ASA) monitors are maintained (including measuring blood pressure every 5 min, continuous EKG, pulse oximetry, and end tidal CO_2_ from a nasal cannula or face massk). Antibiotics (Clindamycin 600–900 mg oy Cefazolin 2–3 g) are used during the implantation procedure. Small doses of anti-anxiety medications, pain medications, and anesthetics can be titrated to allow the patient to be relaxed during the procedure when sedation is required. Using x-ray guidance, two 14-gauge epidural needles are used to induce the loss of resistance technique and access the epidural space (Figure 3c). The leads are navigated in the epidural space, and the correct configurations (i.e., stimulation parameters) necessary to evoke motor potentials are re-tested in the presence of a Medtronic representative. The surgeon makes an incision in the participant’s lower back or their buttock. A pulse generator is then placed in a pocket of tissue between the muscles and the skin. The leads are threaded under the skin to a pocket, and they are connected to a Medtronic Intellis battery (Figure 3). Following hemostasis, the wound is closed in 2–3 layers; dermabond, occlusive dressing, and tape are placed over the wound. A belly band will be provided for patient comfort. Prescription for pain medicine is usually only for three days as this seems to be the most painful part as the single incision heals. Recovery after implantation is usually complete after seven to ten days when the bandages are removed. Every participant is examined three times in the first month for wound checking (one week), dressing change (first week and second week), and reprogramming (fourth week).

d.Spinal segmental mapping

Following both temporary and permanent implantation, participants are scheduled for three to four days of spinal segmental mapping (Figure 4). Spinal mapping is the process of identifying the correct stimulation parameters (frequency, amplitude, and pulse duration) responsible for the activation of the lower extremity muscle groups, correct polarity of the electrodes (cathodes vs. anodes), and the number of the channels needed to evoke the desired muscle contractions and joint movement patterns (hips, knees, and ankles). Stimulation for each specific muscle group and joint per limb are determined using different mapping protocols (Figure 4a). This includes the target mapping of different muscle groups to identify the correct stimulation configurations and parameters. Every effort is made to configure the correct combinations of cathodes and anodes per specific muscle group and joints as well as the correct stimulation parameters (2, 15, and 30 Hz) for the frequency of the pulses and (250–1000 µs) for the pulse duration. We aim to use the minimum amount of current (mA) necessary to evoke muscle contraction and monitor any increase or decrease throughout the trial. From supine position, EMG electrodes will be attached as indicated in our data collection sheet (58). The participant is then asked to make three major movements, such as wiggling the big toe, dorsiflexion of the ankle joint, or moving their entire leg into flexion followed by extension (see Appendix A) with SCES off and then followed by SCES on. We aim to find a multiple-electrode combination (up to 3–4 electrodes per channel) with the correct stimulation parameters. This is followed by two to three days of rest. The participants will be trained on how to use the ES controller to activate the paralyzed lower extremity muscles. For the first 6 months, the target is to achieve overground standing. We will develop recruitment curves for each individual muscle group to attain tonic extensor activity in supine position using three pulse durations (250, 500, and 1000 µs), where the amplitude of the current ranges from 1 to 10 mA with a frequency of 2 Hz. Recruitment curves will be established using 5 different configurations, as shown in Figure 4b. The recruitment curves that yield the highest extensor activity in suping will then be further refined with standing at a frequency of 10–40 Hz to determine the standing configurations. For the following 6 months, we will attempt to determine the configurations that would elicit the rhythmicity in supine lying position as previously highlighted (Figure 4c). These configurations will be tested further during EAW and during stepping and locomotion, if possible using either parallel bars or a standard walker (see Appendix A).

#### 2.3.2. Training

Exoskeleton-assisted walking (EAW)

Prior to training, a member of the research team assists the participant into the device, starting with shoe support (distally) and then going up toward the trunk (proximally) [33,40]. The varying needs of each participant determine how the walking software is adjusted and progressed (Figure 5a). Every effort is made to ensure that all straps are well adjusted but not overly tight to avoid the development of unnecessary episodes of autonomic dysreflexia. EAW is scheduled 3 days/week for 12 months. Initially, training begins in the first-step mode until participants can shift their body weight anterolaterally to achieve stepping. Participants then progress to the pro-step + mode starting with a standard roller walker and then move on to Canadian crutches in approximately 4 weeks. If participants need less assistance, the software is adjusted to the adaptability mode. This mode allows the exoskeleton to gradually lower the assistance provided to the participant as needed based on walking performance. The support ranges from 0 to 100%, where 100% indicates that the unit is providing maximum support and assistance for ambulation. Once the participant is comfortable using Canadian crutches (~4 weeks), we encourage participants to turn on their ES and we start with 100% assistance and drop the support in 5% increments as tolerated until we reach the lowest assistance level possible during EAW [40]. The decision to drop the assistance is based on the participant’s ability to complete 80% (i.e., arbitrary threshold) of the steps in a 10 m distance without cueing. After dropping the assistance, the exoskeleton offers 2.4 s before passively moving the limb using a slow-mode feature. Fortunately, we can differentiate between active (i.e., participant) versus passive (exoskeleton) stepping by setting audible beeps. We hypothesize that the number of unassisted steps will be more than the assisted steps when the SCES is on compared to SCES off. The unit is equipped with two buzzers that help to cue the participants to accurately complete weight shifting prior to stepping. With a trained research assistant providing guidance and support from the back, participants are encouraged to complete 60–90 min of fitting, resting, and walking during a single session. Resting and EAW vital signs (heart, blood pressure, and %SO_2_) will be monitored every 5 min to ensure safety.

b.Task-specific training

Following permanent implantation, the participant is also engaged in 60 min task-specific training 3× per week that focuses on sit-to-stand activities (Figure 5), such as Romanian deadlifts for trunk control using either a standard walker (Figure 5b) or standing frame (Figure 5c). The standing frame facilitates focusing on securing hip extension or volitionally controlling the hip joints in standing position compared to the hip and knee joints when using the walker or the parallel bars. This pattern was randomly alternated on a regular basis. Task-specific training is conducted either before or after exoskeleton training on the same day after a rest period of 60 min. The goal of this training period is to enhance the participant’s ability to enable trunk control and secure independent standing with SCES on. Two or more research assistants are usually involved during the training to provide necessary physical support when needed. Blood pressure is continuously monitored throughout the task-specific training, and participants are provided with 3–5 min of rest between different attempts.

c.Resistance training (EAW + ES + RT group)

Resistance training is performed twice weekly in an open-chain format in the first 12 weeks and followed by another 12 weeks in closed-chain format using the implanted ES. Ankle weights are not applied in the first week to ensure the full knee extension of the lower leg against gravity [20,21]. Weight is increased at an increment of 2 lbs. per week with the criteria that full knee extension should be achieved before a further increase in load [20,21]. Surface NMES is applied to the knee extensor muscles via surface electrodes to induce concentric–eccentric actions. Two 8 × 10 cm^2^ (Uni-Patch, Wabasha, MI, USA) adhesive carbon electrodes are placed on the skin over the knee extensors. The current will be manually increased in 5 s intervals until knee extension is achieved against gravity followed by gradually decreasing the current to induce lengthening actions until full relaxation (i.e., return to the resting position). The current (mA) that causes full knee extension is recorded after each stimulation bout. Each RT session consists of 4 sets of 10 repetitions of NMES-induced knee extensions for 45–60 min. A 5 s/5 s work/rest ratio is used with a 3 min rest between sets, 30 Hz, 450 µs pulses, and a current sufficient to evoke full knee extension [20,21]. Over the additional 12 weeks, each participant will perform closed-chain RT (i.e., a gradual sit-to-stand exercise using a standard walker) using their implanted ES after sitting on an adjustable mat and tuning ES with the following progression under full supervision (Table 3).

During the 6 months of RT, participants are asked to report to the lab 5× per week [3× for EAW + ES and 2× for RT]. Closed-chain RT would result in postural perturbations that activate intrafusal muscle fibers and enhance the sit-to-stand pattern, activating multiple joints and loading the hip and knee extensors.

d.Passive range of motion or passive stretching for the control group (EAW + delayed ES + noRT)

A research assistant sits on a stool, cups the leg proximal to the ankle joint, and moves it from 90 degrees of knee flexion to full extension [58]. The leg is maintained in extension for 5 s and returned to flexion for 5 s. The passive actions are repeated as described in the RT protocol; 10 reps on the right leg followed by 10 reps on left leg for a total of 4 sets × 10 reps. This helps to balance the design between the two groups in the last 6 months of the study. Training is also conducted 5× per week.

#### 2.3.3. Measurements

A.Motor Adaptations

a. The 10 m EAW speed, EAW-unassisted steps, and EMG pattern

Walking time and speed using the 10 m walk test will be determined using either a walker or crutches [40,59]. The procedure will be carried out prior to implantation and every 6 months after implantation. During the 10 MW, the participant performs a walk between two cones set 10 m apart; this is repeated with and without SCES of the hamstrings (HS), medial gastrocnemius (MG), tibialis anterior (TA), vastus lateralis (VL), soleus (SOL), and gluteus medius (GM). The rationale is that the 10 m EAW test improves with training. The test is conducted one time with the ES off and another time with the ES on, allowing each participant to serve as their own control. EMG activity is measured from leg muscles and hip, knee, and ankle joint angles during locomotion (Figure 5). The research team records bilateral EMG (Delsys Inc., Natick, MA, USA) from the soleus, medial gastrocnemius, tibialis anterior, medial hamstrings, quadriceps, and gluteus maximus muscles unilaterally using bipolar surface electrodes with a fixed inter-electrode distance. The ES artifacts will be minimized from the EMG data with a bandpass filter (4th order Butterworth, 20–450 Hz) using custom MATLAB scripts (MATLAB 2015a, The MathWorks Inc., Natick, MA, USA). For baseline and post-intervention data, these filtered and continuous data will then be visually segmented into individual strides using the vertical component of an accelerometer embedded within the EMG sensors. This acceleration signal is lowpass filtered (4th order Butterworth, 3 Hz) using custom MATLAB (Version: 23.2.0.2428915 (R2023b) Update 4) scripts. The data are segmented using the anteroposterior gyroscopic signal embedded within the EMG sensor of the gastrocnemius muscles. Each stride is further segmented into stance and swing phases based on this signal. EMG RMS envelopes are calculated using a moving window of 150 samples for 10 strides. Peak filtered EMG values across the entire stride, each stance phase, and each swing phase is identified and averaged over the total number of strides completed during each assistance level. Using a counter, a researcher allows the subject to ambulate for 10 m and determines the number of EAW-unassisted steps (i.e., without audible cueing) as well as the walking speed.

b. Neurophysiologic assessments

One week prior to intervention (week 1), and during post-interventions 1 and 2, segmental sensorimotor reflexes will be studied by recording H-reflexes and nociceptive reflexes. In the same spinal segment (but regarding a different pool of motoneurons), spinal motoneuron excitability is assessed by recording F-waves [60,61]. The soleus H-reflex is recorded (20 Hz–20 kHz bandpass) with a surface EMG electrode on the motor point of the soleus muscle with the subject in a seated posture. Electrical stimulation (1 ms duration pulses) is applied using a constant current stimulator (DS7AH, Digitimer Ltd.) at 0.5 Hz to the tibial nerve at the popliteal fossa by a monopolar electrode. Stimuli will be delivered at gradually increasing intensities to obtain the H-reflex of the maximal amplitude (H-max). The amplitude and latency of H-max are measured. The nociceptive flexion reflex is recorded (20 Hz–20 kHz bandpass) with surface EMG electrodes on the motor point of the biceps femoris muscle. Electrical stimulation (a train of five 1 ms duration pulses with an inner frequency of 300 Hz) is applied to the sural nerve at the ankle by a bipolar electrode. A nociceptive reflex threshold will be determined by delivering stimuli at gradually increasing and decreasing intensities until a stable reflex response is obtained with a latency ranging between 80 and 180 ms. Three stimuli are delivered at 20% above the reflex threshold. The stimulation of motor F-waves to tibial nerve is recorded (20 Hz–20 kHz bandpass) with a surface EMG electrode on the motor point of the abductor hallucis muscle. Electrical stimulation (0.2 ms duration pulses) is applied at 0.5 Hz to the tibial nerve at the ankle with a bipolar electrode, while the abductor hallucis muscle is at rest. The intensity at which F-waves are clearly identifiable (F-wave threshold) and the intensity at which the M-wave of the maximal amplitude is obtained (M-max threshold) are recorded. Ten stimuli are delivered with an intensity set at approximately 20% above the M-max threshold [35,36].

c. Overground ambulation for both groups

During post-interventions 1 and 2, overground ambulation without wearing the exoskeleton is tested using a zero-gravity gait system (Bioness System) during the 10 m walking distance with a standard walker equipped with sensors to determine the exact weight bearing on the arms. We will also determine the number of unassisted steps and the speed of walking.

d. Peak torque

Peak torque for both groups is evaluated using a Biodex isokinetic dynamometer (Shirely, NY) [20,41,62]. Participants are seated with both the trunk–thigh angle and the knee–thigh angle at 90°. After transferring using a ceiling lift, each participant is securely strapped to the test chair with a crossover shoulder harnesses and a belt across the hip joint. The axis of the dynamometer is aligned to the anatomical knee axis, and the lever arm is attached 2–3 cm above the lateral malleolus. Before measuring isometric torque, the passive tension of the right knee extensor muscle group is measured at 5, 30, 60, 90, 180, and 270 degrees/sec as an index of spasticity. Isometric torque is measured using current amplitudes of 50 and 100 mA at a frequency of 30 Hz and pulse duration 450 µs. Isokinetic torque is measured at 60, 90, and 180 degrees/sec using the same stimulation protocol. Following implantation in post-measurements 1 and 2, each participant is initially asked to kick their leg as strongly as possible to measure the torque generated at the knee joint when the ES is on. Each condition was tested with ES on and off to compare the effects on torque generation and spasticity [62]. This provides insight into the ability to activate the knee extensors and the force generated with ES in persons with motor-complete SCI.

e. Walking Index for Spinal Cord Injury II (WISCI II)

The Walking Index for Spinal Cord Injury Scale (WISCI 2) has demonstrated a response to change in the Spinal Cord Injury Locomotion Trial and in clinical validation studies [58,63]. When scoring the WISCI, the examiner checks the descriptors that apply to current walking performance and then assigns the highest level of walking performance. In scoring one level, the level at which the participant is safe as judged by the therapist should be chosen, with the participant’s comfort level described. If devices other than those stated in the standard definitions are used, they should be documented as descriptors. The participant is observed by trained personnel, and the WISCI level is recorded on a scale of 0 to 20 at baseline and post-measurement 2 (baseline and post-measurement 2 WISCI). The participant is observed again at the defined interval (Interval WISCI). The change in score is calculated by subtracting the baseline WISCI from the Interval WISCI, which equals the change in WISCI (Changed WISCI).

B.Cardio-metabolic and Autonomic Adaptations

f. Blood pressure and oxygen uptake (VO_2_)

One week prior to intervention (week 1) post-measurements 1 and 2, VO_2_ is measured using a COSMED K4b2 (COSMED USA, Chicago, IL, USA) portable metabolic unit [33,64]. While wearing the robotic exoskeleton, the participant is asked to walk for 6 min for the measurement of the walking speed and distance (6MWD). The test begins with the participant sitting with the exoskeleton for 5 min, followed by a 3 min standing phase, after which the participant performs a 6 min walk test, a 3 min standing phase, and finally a 5 min sitting phase [33,64]. After calibration, subjects are asked to wear a mask to monitor VO_2_ and carbon dioxide (VCO_2_) production. A three-minute resting phase allows the subject to learn to tolerate breathing with the mask on, while they are wearing the exoskeleton suit both sitting and standing (3 min each). VO_2_ and VCO_2_ are monitored throughout EAW for 6 min to determine total energy expenditure using the Weir equation. After EAW, VO_2_ is measured during the five-minute recovery phase [33,64]. The five minutes of recovery are recorded to determine the impact of each intervention on energy expenditure and substrate utilization. The heart rate (via polar HR monitor) and blood pressure (COSMED 740) are used to monitor changes during resting, exercise, and recovery periods. An HR monitor is placed on the subject’s chest below the xyphoid process, and it remains there throughout the study. HR is recorded every 30 s throughout the training session. Blood pressure is recorded continuously before exercise, every 2 min during EAW, and for another 5 min during the recovery phase.

g. Body mass index (BMI)

After voiding, the participant is propelled onto a wheelchair-weighing scale. After weighing the participant and their wheelchair (kg^−1^), they are assisted in transferring to an adjustable mat and their wheelchair is weighed by itself (kg^−2^) [20,21]. The weight of each participant is calculated by subtracting (kg^−2^) from (kg^−1^). The height is determined while the participant is lying in supine position. After transferring to the mat, the researcher helps align each participant properly. Two smooth wooden boards will be placed at the participant’s head and heels, and the distance between them corresponds to the height to the nearest cm unit. The BMI (kg/m^2^) is calculated as weight (kg)/height^2^ (m^2^) [65,66,67].

h. Dual energy x-ray absorptiometry (DXA)

DXA will be used to measure body composition, specifically regional and total lean mass. Total body and regional DXA scans will be performed using an iLunar DXA (Lunar Inc., Madison, WI, USA) bone densitometer at the Richmond Veteran Affairs Medical Center in Richmond, VA, USA. After 20 min of rest on a padded table, the scans are performed and analyzed by a trained certified DXA operator using the iLunar system. Total and regional lean mass are determined using total and regional DXA software (enCORE Version 16) [65]. Using the equations developed by Lester et al., we will predict muscle CSA at baseline and post-measurements 1 and 2 [66]. Muscle Mass _MRI-WM_ = 0.952 (LM_DXA_) − 0.073 (*r*^2^ = 0.90, SEE = 0.23 kg, *p* < 0.0001). A second equation has been developed to predict the MRI absolute thigh lean mass, after accounting for intramuscular fat and bone mass from DXA LM: MM_MRI-ABS_ = 0.824 (LM_DXA_) + 0.018 (*r*^2^ = 0.82, SEE = 0.28 kg, *p* < 0.0001) [45]. The two equations should allow for predictions of the changes that occur in muscle CSA without the need to use magnetic resonance imaging.

i. Basal metabolic rate (BMR) using indirect calorimetry

After an overnight fast for 10–12 h, participants are kept in a dark room for 20 min to attain a resting state during which the basal metabolic rate (BMR) is measured using a canopy that covered the whole head and portable COSMED. The BMR and respiratory exchange ratio are recorded [67].

j. Blood lipids, serum total testosterone, brain-derived neurotrophic factors (BDNF) and IGF-1, inflammatory biomarkers, and oral glucose tolerance test (OGTT)

Following the BMR testing, fasting lipid profiles (HDL-C, LDL-C, total cholesterol, and TG) are assessed as 10 mL of blood is collected from the indwelling venous catheter, and lipids are determined using standard analyses procedures [67]. Measurements of total testosterone are performed by radioimmunoassay after sample extraction and column chromatography [21]. The intra-assay coefficient of variation (CV) is 12.5% or less for all quality control samples analyzed with the study samples. Plasma IGF-I, IGFBP-3, and brain-derived neurotrophic factor (BDNF) concentrations are measured with an immunoluminometric assay (Quest Diagnostics, Madison, NJ, USA) and RIA (Diagnostics Systems Laboratories Inc., Webster, TX, USA), respectively. The intra-assay precision of IGF-1 is 4.6% at 50 ng/mL and 3.6% at 168 ng/mL [67]. Inflammatory biomarkers (CRP, IL-6, and TNF-α) are also determined with standard analyses procedures using available enzyme-linked immunosorbent assay (ELISA) kits [67]. For the oral glucose tolerance test (OGTT), participants drink 75 g of glucose solution and then have their blood drawn at 30 60, 90, 120, 150, and 180 min after glucose consumption [67]. Duplicate blood samples are taken; the plasma is separated from the cells, placed in tubes containing 10% EDTA, and frozen at −70 °C until analysis. Plasma glucose is measured with the autoanalyzer glucose oxidase method, and the plasma insulin concentration is determined by commercial radioimmunoassay using single-antibody kits at the pathology laboratory.

k. Near-infrared spectroscopy (NIRS)

The amount of oxygenation (O_2_Hb) and deoxygenation (HHb) in the gastrocnemius muscle is determined prior to and at the end of the 6 and 12 months of training using a portable NIRS unit (Portamon, Artinis Medical Systems) [68]. An NIRS probe is placed longitudinally along the belly of the muscle of the right leg approximately 3–5 cm below the popliteal fossa. The probe is secured with a Velcro strap around the calf. A blood pressure cuff (Hokanson SC-10D, Hokanson, Inc., Bellevue, WA, USA) is placed proximal to the NIRS probe as high as possible above the knee joint. The blood pressure cuff is controlled with a rapid-inflation system (Hokanson E20, Hokanson) set to a pressure of 250 mmHg and powered with a 15-gallon air compressor. Resting muscle oxygen consumption is first measured through the inflation of the cuff for 30 s. Ischemic calibration, also known as a physiological calibration, is performed to normalize the NIRS signals for each test. It provides a reference point for 0% oxygen saturation in the muscle during a period of arterial occlusion (ischemia) and a reference point for 100% oxygen saturation when the cuff is released. Muscle oxygen consumption (mVO_2_) is determined using the rate of change (i.e., slope) of the HHb signal during walking compared to the baseline sitting position. Measurements are conducted during the 6 min walk test in conjunction with wearing the mask to measure energy expenditure. The test is conducted in conjunction with measuring peak oxygen uptake during EAW.

l. Tilt table test for monitoring blood pressure

Autonomic function will be assessed with standardized procedures that are routinely used for persons with SCI in a research setting [62,69,70,71]. Heart rate variability is an indirect index of the ability of the autonomic nervous system to adapt to different stressors. Heart rate variability is calculated offline via computer software (LabChart, AD Instruments) (Version 8.1.19) that utilizes data downloaded from a 5-lead ECG. Participants are asked to lay quietly in supine position and rest for 5 min while a 5-lead ECG collects beat-by-beat heart rate data and a finger cuff collects beat-by-beat continuous blood pressure (Figure 6). The tilt table is then gradually raised to an angle of 60 degrees. Straps across the participant’s knees, hips, and chest ensure that the participant stays upright while the table is tilted. This position is maintained for up to 10 min while the heart rate and blood pressure are monitored. The table is lowered back to supine before 10 min if the participant’s systolic blood pressure drops by more than 40 mmHg from resting baseline, or upon the request of the participant if orthostatic symptoms become too severe, as per standard methods [62]. The testing concludes with another 5 min period of supine rest while the heart rate and blood pressure are continuously recorded. This is followed by repeating the same protocol with SCES on after a 30 min washout period. The SCES is turned on 3–5 min in supine lying position before performing the head-up tilt maneuver following the above sequence [62].

The data collected from these periods are used offline to calculate heart rate variability and the general autonomic response to a change in posture. The LabChart software (Version 8.1.19) has built-in modules that are used to calculate heart rate variability from the recorded ECG data.

The following outcomes will be derived from the proposed tilt table test procedures:-Duration of tilt: The length of time the participant can safely maintain tilt. It is anticipated there will be a ceiling effect on this outcome as some participants may safely tolerate the entire 10 min duration of the tilt from the first assessment.-Low-frequency (LF) power during tilt: The frequency spectrum power of low-frequency oscillations in RR intervals derived from ECG recordings. The spectral power itself will be determined offline in our data collection software (LabChart) using built-in calculation modules. LF power is indicative of how much baroreceptor activity underlies changes seen in the heart rate or blood pressure when changing postures.-High-frequency (HF) power during tilt: The frequency spectrum power of high-frequency oscillations in RR intervals derived from ECG recordings. The spectral power itself will be determined offline in our data collection software (LabChart) using built-in calculation modules. HF power is indicative of how much parasympathetic activity underlies changes seen in the heart rate or blood pressure when changing postures.-Cardiovascular responses to initiation of tilt: The 30 s average of HR, SBP, and DBP from the beginning of tilt minus the 30 s average from the end of the supine baseline.-Cardiovascular responses during tilt: The 30 s average of HR, SBP, and DBP from the beginning of tilt minus the 30 s average from end of tilt.

C.Urodynamic Measurements

Urodynamics are tested to determine the effect of either intervention on bladder functions [72,73,74,75]. Previously, locomotor training with ES has been shown to improve bladder functions in persons with SCI [51]. Following implantation, participants report for 2–3 visits to map the best cathodal and anodal combinations as well as the stimulation parameters for testing bladder functions. Multiple urodynamic parameters are used to demonstrate ES-induced volitional control at baseline and at post-measurements 1 and 2. For this study, only individuals with motor-complete SCI managing their bladders with clean intermittent self-catheterization are recruited. Prior to the initiation of treatment a 3 day frequency–volume chart is created. This provides baseline information on bladder capacity. When participants present for baseline urodynamics, in order to standardize fill rates, the initial fill rate will be set to 10% of the maximum catheterized volume (reported in 3-day void diary) per minute. This ensures that all bladders are filled at a proportionally equivalent rate. The cystometric capacity is then defined as the volume at which involuntary contraction with leakage occurs during this first fill, and 10% of this cystometric capacity is used for any subsequent fills in which urodynamic data are collected [72,73,74,75].

The following urodynamic outcome measures are evaluated:Development of volitional voiding:

In individuals with motor-complete SCI and baseline urodynamics confirming complete involuntary detrusor contractions, we evaluate the presence of volitional voiding defined as the presence of any voluntary voiding after the “permission to void” (PTV) command is given.

2.Time to volitional voiding:

We measure the time from PTV to the onset of voluntary voiding.

3.Voiding efficiency:

This is defined as (voided volume)/(infused volume + post-void residual) and is a standard measure of voiding efficiency.

4.Suppression of involuntary contractions:

In participants with involuntary detrusor contractions, we measure the change in peak pressure of the involuntary contractions to quantify the degree of the suppression of involuntary contractions.

5.Suppression of urinary urgency:

If participants report urinary urgency, the ability (yes or no) to suppress involuntary bladder contractions is assessed and compared.

6.Alterations of bladder sensation:

The international continence society (ICS) recommends the use of standard verbal sensory thresholds, including first sensation of filling, first desire to void, and strong desire to void during urodynamics. In this study, we compare the volume at which these verbal sensory thresholds occur at baseline and after ES.

7.Assessment of bladder outlet obstruction:

The ICS nomogram and the Bladder Outlet Obstruction index (PdetQmax—2Qmax) are used to compare BOO pre vs. post ES treatment.

8.Bladder contractility:

Bladder contractility is assessed using the bladder contractility index defined as PdetQmax + 5Qmax. We have chosen to measure multiple outcome variables because the aim is exploratory in its nature, and we would like to determine what outcome variables are likely to be sensitive to this intervention.

## 3. Recruitment Strategy

We plan to have four participants recruited yearly to meet our sample size. Approximately one to two subjects will be screened every other month. They will then be randomized to either group. Table 4 highlights the number of participants enrolled, completed, or withdrawn from the current trial.

## 4. Statistical Analyses

Means and standard deviations or frequencies and percentages, for all study variables will be calculated at all timepoints. T-tests and Pearson chi-square tests will be used to compare characteristics that may act as confounding variables, including age, level of injury, and time since injury.

Primary analyses will be mainly descriptive given the pilot nature of this study. Additionally, means and standard deviations will be estimated for utilization (the number of sessions completed) and each outcome. Lastly, an analysis of covariance (ANCOVA) will be used to test for differences between groups post interventions for each outcome using the baseline score as a covariate. If the normality of the groups appears to be violated in the sample, normalizing transformations or non-parametric statistics will be considered.

Within-group, within-timepoint comparisons of all outcomes with ES versus without ES will be used for these comparisons to test the acute effects of ES at each timepoint, such as T-tests (or a Wilcoxon test if the final data are non-normal). Within-group, between-timepoint comparisons of all outcomes with ES versus without ES will be carried out to test the effects of training and the use of ES over time. A mixed ANOVA will be used for this comparison to allow testing for condition (ES vs. no ES) × time interaction testing, with *p*-value corrections for repeated measures as appropriate. Lastly, between-group comparisons, at P1, between-group (6 months of EAW + ES) versus 6 months of EAW alone), and between-condition (ES on vs. ES off) comparisons will also be conducted. These will determine whether the difference in the ES on vs. ES off outcomes is differentially affected by 6 months of EAW training with ES versus 6 months of EAW training alone. At P2, between-group (6 months of EAW + ES concurrent with 3 months of NMES-RT and 3 months of sit-to-stand training versus 6 months of EAW + ES) and between-condition (ES on vs. ES off) comparisons of all outcomes will be carried out with mixed ANOVA. This will determine whether the addition of 3 months of NMES-RT followed by 3 months of sit-to-stand training differentially affects the outcomes compared to only 6 months of EAW training with ES.

Additionally, the following statistical comparisons can be made within-group, within-timepoint, between-timepoint, and between-group for the data collected from the tilt table test.

In addition to these statistical tests, there are seven distinct categories of heart rate and blood pressure response to postural changes that have been identified in persons with SCI [44,45]. Using these published response categories, we will be able to determine (1) which response category each of our participants falls into and (2) determine if the intervention changes the response category into which a participant falls and whether their responses are closer to those of a healthy, non-injured individual.

## 5. Discussion

The current trial implements the use of percutaneous ES for restoring volitional motor control, as well as improving cardiovascular and metabolic profiles, body composition, bladder control, and overall quality of life, in persons with SCI through a 12-month rehabilitation period of EAW + ES + RT compared to EAW + delayed ES + no RT. The design of the trial is based on the rationale to enhance motor outcomes via the implantation of percutaneous SCES and the later addition of NMES-RT. We are hopeful that the successful completion of this procedure will decrease the overall seated time in wheelchairs for participants. Seated time has been shown to be an independent risk factor for mortality in the general population [76]. Furthermore, EAW is used for locomotion training as an alternative method to BWSTT due to the ambulation benefits it can provide at a low metabolic cost to the participant [37,43]. EAW has demonstrated an ability to decrease spasticity, improve bowel movements, and improve quality of life in persons with SCI [77,78]. Unlike when using EAW with ES [62], researchers in one study found that 24 weeks of EAW did not affect any symptoms of spasticity in individuals with SCI [37].

### 5.1. Novelty of the Current Work

Compared to previous work [22,23,24,25,26,27,28,29], we propose to use percutaneous leads compared to the paddle implantation [41,54,57,62,79,80,81]. The latter requires laminectomy to accurately place the paddle in the epidural stimulation to achieve target segmental stimulation [22,23,24,25,26,27,28,29]. Despite the potential complications that may arise from laminectomy [82,83], previous trials have successfully demonstrated achievements in motor outcomes similar to standing, stepping, and overground locomotion [22,23,24,25,26,27,28,29]. Percutaneous leads offer the capability of steering the epidural space without the need to perform a laminectomy [80,81]. Additionally, others have successfully indicated the potential of percutaneous SCES to achieve specific target stimulation of the desired motor segments [79]. The use of percutaneous ES is considered a safe and feasible neuromodulation approach and capable of covering several spinal cord segments through the placement of the leads in a staggering approach when deemed necessary, rather than a simple parallel, even approach.

Another novelty is the use of EAW as a procedure for locomotor training following implantation. Previous work has established the capability of ES to enhance walking outcomes in persons with SCI when paired with EAW [40,41,62]. ES and EAW together resulted in increased muscle activation and decreased assistance provided by the exoskeleton [40]. Another report indicated that one participant was able to achieve independent standing with minimal assistance and independent stepping [41]. Other researchers found that EAW + transspinal stimulation (TS) resulted in continuous stepping with improvements in coordination, autonomic function, including cardiovascular and thermoregulatory profiles and lower urinary tract infection [47]. Spinal cord ES in combination with LT provides sensorimotor training of the spinal circuitry and facilitates standing stepping using a multi-modal approach to enhance neuroplasticity [22,23,24,25,26,27,28,29,30]. A recent case report demonstrated the possibility of creating a close-loop integration between EAW and SCES to achieve spatiotemporal stimulation in a T12 person with complete SCI [84]. Furthermore, we noted that interleaving configurations could complement each other to enhance the results of stepping [81]. Different configurations of ES were tested separately and then together and then interleaved to successfully initiate 16 consecutive steps [81]. In the current study, spinal cord ES will be delivered at a sub-motor threshold to ensure the neuromodulation of the spinal cord locomotor centers during EAW [41]. This is monitored through the ability of the person to initiate stepping and the decrease in the swing assistance provided by the exoskeleton [40,41]. The current study will also provide evidence regarding the effects of spinal cord ES on cardio-metabolic risk factors and body composition parameters in persons with SCI [85].

### 5.2. Autonomic and Cardio-Metabolic Benefits

Lastly, in addition to its sensorimotor benefits, ES is recognized for enhancing autonomic regulations in persons with SCI [62,69,70,71]. Thus far, in our trial, three participants with C5-T4 SCI (AIS A and B) were implanted with a 16-electrode array to stimulate lumbo-sacral enlargement. Following the orthostatic challenge, ES (15–30 Hz) managed to ameliorate the drop-in blood pressure (BP) that occurred with the stimulator off [69]. During standing, systolic and diastolic BP increased by 27–80% after turning the ES on. Another study reported improvement in systolic BP following orthostatic challenge after using ES in four individuals with motor-complete tetraplegia [70]. The effect persisted after training because of the possible activation of sympathetic vasomotor efferent and improved venous return [69,70]. In a previous case report, our lab noted that percutaneous ES configurations intended to enhance walking function resulted in improvement autonomic regulation in response to orthostatic challenge in a C7 AIS person with SCI [62]. Activating ES in a 5 mA stabilized participant resulted in systolic blood pressure at an average of 111.5 mmHg, above levels reached without ES, and also produced the highest low-frequency and high-frequency components of heart rate variability analysis [62]. Figure 6 highlights a step-by-step procedure on how we measure fluctuation in blood pressure in a T4 AIS A person with SCI in response to orthostatic challenges. It is interesting to note that SCES application may lead to the development of autonomic dysreflexia that is characterized by a sudden increase in blood pressure above resting baseline measurements [86]. A recent report demonstrated that implanting percutaneous leads resulted in a 22% incidence of the development of AD with participants at rest [86]. However, 97% of all the episodes were asymptomatic [86]. Therefore, monitoring blood pressure is an essential aspect of any SCES trial. Additionally, the modulation of the stimulation parameters similarly to the amplitude of the current or pulse duration in a supine resting position may likely reduce the incidence of the development of AD in response to SCES.

### 5.3. Percutaneous ES Configurations

Several trials have attempted the process of configuring ES to enhance motor activities. Today, this process varies greatly among different centers that utilize this approach [30,31]. This process highlights the discrepancy in either enabling the lumbosacral circuits and pursuing high-intensity task-specific training or directly stimulating the spinal locomotor center in a spatiotemporal manner with different phases of the gait cycle [23,32]. The latter process seems to successfully restore motor function within 1–3 days post implantation [32]. However, its long-term applicability in real-life scenarios or activities of daily living has yet to be established [23].

Applications of percutaneous SCES have limited information regarding successful configurations of the cathode–anodes or stimulation parameters (pulse durations, frequency, and amplitude of the current) to achieve desired motor function [54,57]. This has led our research team to adopt several techniques to correctly configure percutaneous ES. We primarily relied on developing recruitment curves at 2 Hz and at an amplitude ranged from 1 to 14 mA using three different pulse durations (250, 500, or 1000 µs) in supine lying position [41,54,57]. We limited the selection to five spatial configurations (two wide-field and three narrow-field configurations) [54]. We then relied on the visual inspection of the EMG activities that are likely to elicit greater extensor patterns relative to the flexors and tested these configurations with a standing frame at frequencies 10–60 Hz while a member of the research staff provided hip joint support [41,54]. The visual inspection of the recruitment curves or EMG activities was subjectively considered and relied primarily on the experience of the examiner. This has led our research team to retrospectively analyze the recruitment curves and fit them in a sigmoidal function and calculate the strength of the slope of the curve using the R^2^-value [54]. The calculated R^2^s of the extensor–flexor curves were then used to determine the peak slope ratio. A ratio close to 1 or greater was then used to highlight the necessary configurations to achieve standing. This technique has limited the number of subjective selections of the configurations and provided the possibility to specifically target the configurations of agonists–antagonists for different joints (hips, knees, and ankles) to achieve standing after SCI [54].

On the other hand, we have shown that deciphering configurations that lead to inducing the rhythmicity of the inter-neuronal circuitries of the lumbosacral segments (i.e., central pattern generator) is likely to enhance exoskeletal performance, enhance non-functional stepping between parallel bars, and induce overground stepping with an assistive device [64]. In a case report, we noted that rhythmic configurations can be interleaved to reduce adductor muscle tone and enhance overground stepping in a person with tetraplegia [81].

## 6. Limitations

One of the most feared complications in using SCS percutaneous lead implants is the risk of lead migration when compared to surgical paddle implantation [41,57,81]. The migration of the leads directly impacts the effects of stimulation on training and results in non-specific stimulation as confirmed by MRI reconstruction of the lumbosacral segment [57]. If ES is not applied over the targeted spinal cord segments, then this would result in non-specific stimulation or absence of the stimulation to the desired muscle groups, which hinders the overall outcomes of the restoration of motor control in persons with SCI. While modern anchors and anchoring techniques have greatly improved, more work can be done in this area as lead migration may result in loss of functional gains in persons with SCI. 

Overground locomotor training depends on each individual’s compliance, motivation, and tolerance for the demands of EAW, ES, and RT. While this study focuses on individualized adjustments based on real-time feedback from the participants, the outcomes may vary due to differences in response to stimulation and resistance training.

Furthermore, the inclusion of AIS A and B participants with motor impairments between C5 and T10 could affect outcomes, as injuries above C5 are known to affect upper limb function, which may hinder the operation of the exoskeleton, and injury levels below T10 may have lower motor neuron injury. However, this may be mitigated by carefully stratifying participants based on injury level and conducting detailed neurological assessments using ISNCSCI standards to ensure uniformity in the study population.

Another limitation is the challenge of transportation and travel for participants. Due to transportation barriers, persons with SCI have difficulty commuting to the research center. Even though allowances are set aside to accommodate transporting participants using a VA van for study measurements, lodging proves to be an issue, which requires more funding to alleviate the burden for participants traveling from distant locations entirely. This is further compounded by the fact that this study is being conducted at a single center, which may help reduce the variability in the performance of the protocol; it reduces the geographical reach and may result in limiting the diversity of the sample size. To address this limitation, a more extensive SCI registry may need to be used. Additionally, future studies may consider shorter training duration than the current trial as well as providing instructional training in a home-based environment. 

The participant’s height and weight can also limit participation as the individual needs to weigh less than 220 lbs and be within a height range of 5′2–6′3, which is also a key criterion for using the exoskeleton in this study, based on the (recommendations of the manufacture of the exoskeleton brand). Furthermore, the participant’s body weight may affect their ability to fully engage in the EAW training and affect the battery life of the operating exoskeleton and subsequent motor performance. Even though diet is critical in maintaining muscle mass and overall health, this study can only partially control or account for participants’ dietary intake outside the study environment. Dietary variations could influence muscle hypertrophy, metabolic outcomes, and training effectiveness.

## 7. Trial Status

It is currently an active trial that has been open for enrollment since (14 October 2020) with an anticipated completion date of (25 August 2025). This study was approved by (Richmond VA IRB since 10 May 2020 and R&D since 14 October 2020). Recruitment started on (14 October 2020) and will end on (25 August 2025).

## Figures and Tables

**Figure 1 jcm-14-01829-f001:**
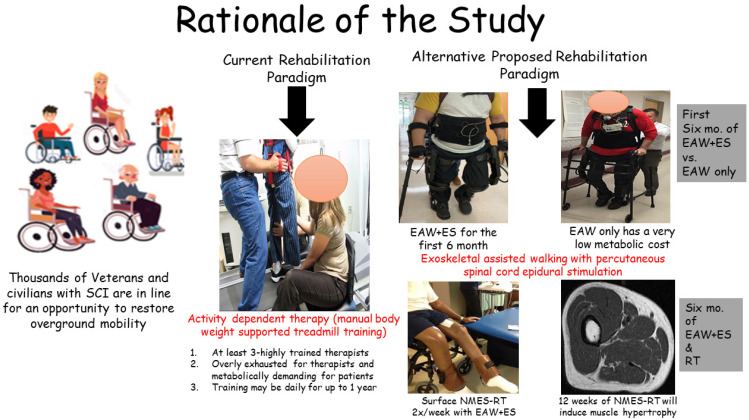
The rationale of this study. The current rehabilitation paradigm consists of BWSTT + ES and is fatiguing for participants and research staff. Our proposed alternative utilizes EAW + ES + RT, which should maximize participant improvement and allow for more efficient use of resources.

**Figure 2 jcm-14-01829-f002:**
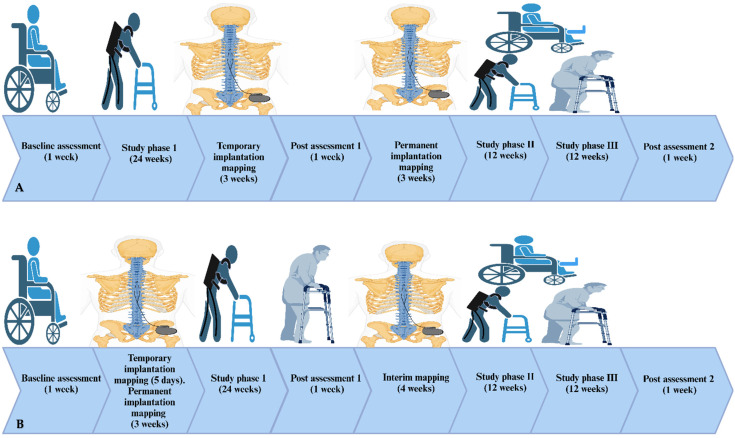
Representative image of the timeline of the two study arms. After screening and consent, participants ae randomized into either the (**A**) exoskeletal-assisted walking (EAW) + SCES + resistance training (RT) group or (**B**) EAW + 6 months delayed SCES + no RT group. Each participant undergoes an intervention of approximately 12 months. Measurements are carried out at 6-month intervals. Following baseline measurement, temporary implantation is performed, followed by 3–5 days of mapping. This is followed by permanent implantation and by approximately 3 weeks of mapping. The EAW + ES + RT group undergoes temporary implantation followed by permanent implantation during the first 6 months of the study. RT is then added during the second 6 months of the study. During the two phases of the study, participants are instructed to conduct 60 min of task-specific training. The EAW + delayed ES+ no RT group undergoes the first 6 months of EAW only. This is followed by the implantation of ES at the beginning of the second 6 months with passive movement training that is conducted twice weekly. (Created in BioRender). https://BioRender.com/b15f572 (accessed on 30 December 2024)).

**Figure 3 jcm-14-01829-f003:**
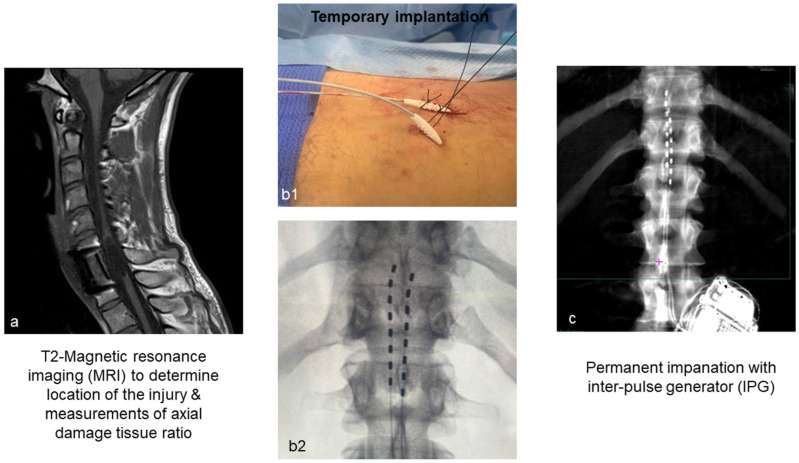
The figure summarizes step-by-step procedures of different phases of implantation. (**a**) The process starts with conducting T2-MRI to verify the location and the size of the injury as well as to determine the axial tissue damage, as previously highlighted. (**b1**,**b2**) highlight the procedure of temporary implantation. The procedure is conducted in a minor procedure room, and after local anesthesia, the temporary leads are threaded in the epidural spaces and then anchored from outside the body once the anatomical location is verified by fluoroscopy as indicated in (**b2**). (**c**) A representative DXA image of the spinal region depicting lead placement with an inter-pulse generator. The eight rectangular blocks on the leads are representative of the electrodes. The leads are seen to be staggered, where the left lead covers the midline of the vertebral body of T11 to the upper border of the L1 vertebral body, and the right lead extends from the T11-T12 intervertebral disk to the middle portion of the L1 vertebra.

**Figure 4 jcm-14-01829-f004:**
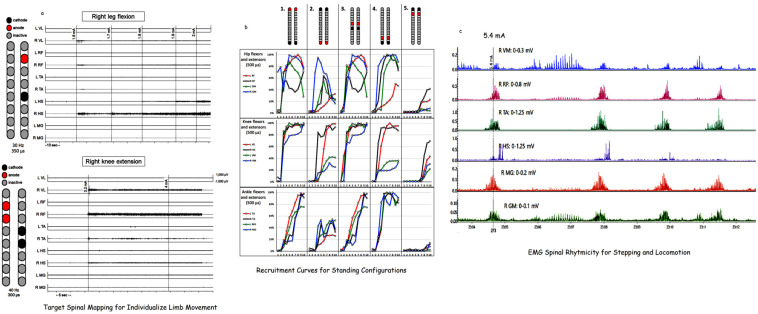
Spinal mapping procedures across the entire trial. (**a**) Right leg flexion and right knee extension. Epidural stimulation configuration for one participant resulting in right leg flexion and right knee extension. All EMGs are shown on the same scale. Tonic motor activity (right leg flexion: primarily R HS) and (right knee extension: extensor muscles of R VL and R RF) occur at a stimulation amplitude of 1.6 mA and 3.2 mA, respectively. (**b**) Evoked potential curves from spinal cord stimulation with different configurations. Five different epidural stimulation configurations are used (red = anode, black = cathode, gray = inactive) to stimulate the lumbosacral spinal cord of one participant at 2 Hz, with stimulation amplitudes ranging from 1 mA to 10 mA. EMG amplitudes from recorded muscles are normalized to the maximal amplitude obtained during the recording session. These curves show the research team whether “wide-field” configurations (1 and 2) or “narrow-field” configurations (3, 4, and 5) lead to the best individual and multi-muscle activity patterns for different functions. Testing multiple configurations at different pulse widths (250, 500, and 1000 µs) will give the team more insight into what stimulation parameters and locations can be used to more precisely target desired muscles to achieve standing. (**c**) Rhythmic EMG activities of different muscle groups after setting the amplitude at 5.4 mA and 2 Hz are likely to be used in practicing stepping and locomotion at higher frequencies during either EAW or overground locomotion. Hz = hertz (stimulation frequency); µs = microseconds; L = left; R = right; VL = vastus lateralis; RF = rectus femoris; TA = tibialis anterior; HS = hamstrings; MG = medial gastrocnemius; sec = seconds; µv = microvolts.

**Figure 5 jcm-14-01829-f005:**
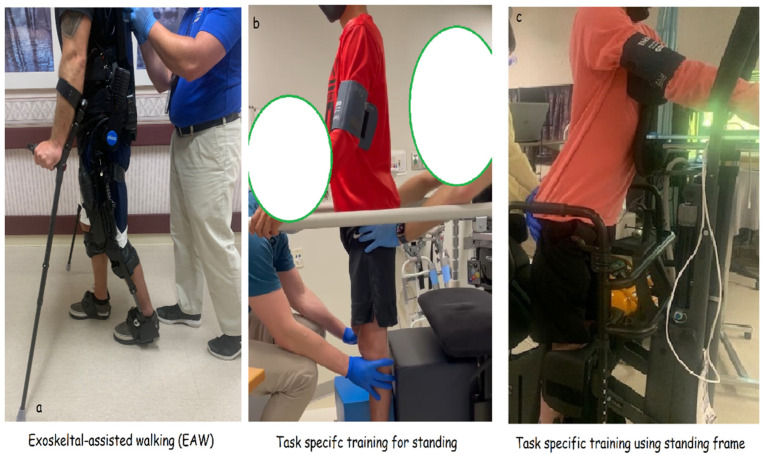
(**a**) Representative image of participant during exoskeleton-assisted walking (EAW) with SCES on. The physical parameters of the exoskeleton are adjusted for each participant. The assistance level provided by the exoskeleton is tested and adapted based on individual’s performance. (**b**) Task-specific training is performed with SCES on in an attempt to enhance overground standing. (**c**) Task-specific training using a standing frame to lock the knee joints and focusing on volitional control of the hip joints.

**Figure 6 jcm-14-01829-f006:**
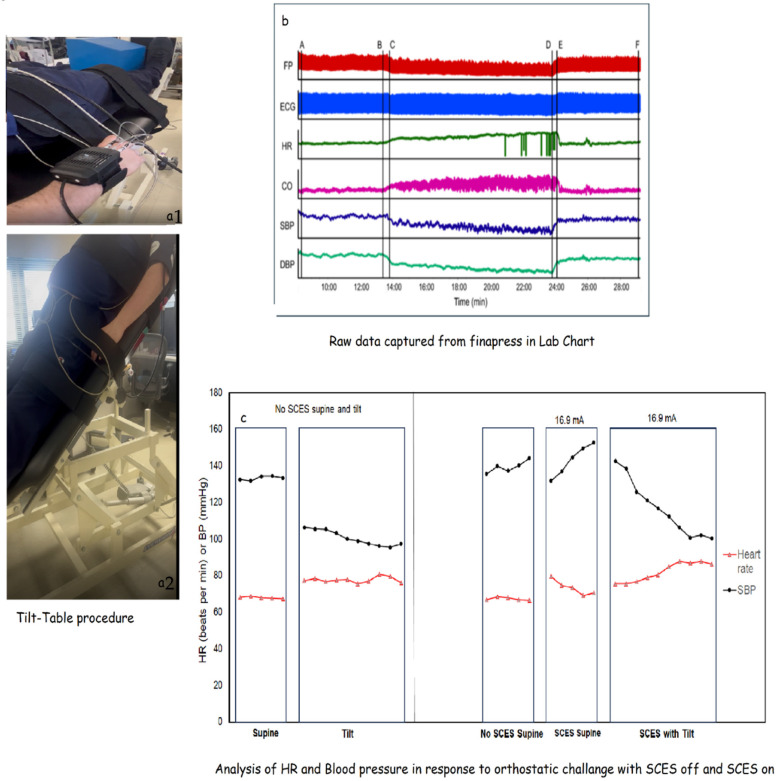
Representative image of autonomic response during a 60-degree head-up tilt test. (**a1**) highlights the procedure of placement of the finapres (FP) sensor to measure blood pressure as well as brachial pressure, and this is followed by calibration. (**a2**) A motorized tilt table that is gradually adjusted from supine position to 60 deg head tilt using a table goniometer. (**b**) Raw data are captured in supine position, tilting position, and recovery to supine position. (**c**) Initially, an assessment is carried out during a testing point with no SCES delivered. The test requires the participant to be in supine lying position for 5 min, which is followed by an attempt to perform a head-up tilt utilizing a tilting table for 10 min. This is followed by a 5 min recovery phase in supine position. The same protocol is repeated with SCES on after a 30 min washout period. The SCES would be turned on for 3–5 min in supine lying position before performing the head-up tilt maneuver. The timepoints for the test are marked by solid black vertical lines. Finapres (FP), electrocardiogram (ECG), heart rate (HR), cardiac output (CO), systolic blood pressure (SBP), and diastolic blood pressure (DBP) are the variables measured on the vertical axis.

**Table 1 jcm-14-01829-t001:** Case reports of locomotor effects of ES.

Researcher	Subjects	Management	Outcome
Harkema [26]	Chronic Paraplegia	ES over L5-S1 7 months of BWSTT	○ Improved voluntary movement ○ Improved EMG pattern
Huang [31]	C5-6 AIS C Tetraplegia T8 AIS Paraplegia	ES over T10-L2 with BWSTT	○ Increased walking distance and duration ○ Improved EMG pattern
Grahn [32]	T6 AIS A Paraplegia	ES over L5-S1 2 weeks of supported movement	○ Increased volitional motor control ○ Stepping
Rejc [33]	C7 AIS A	ES over L5-S1	○ Activated proximal leg muscles ○ Standing without ES

ES: epidural stimulation; BWSTT: body weight supported treadmill training; EMG: Electromyography; AIS A: ASIA impairment scale.

**Table 2 jcm-14-01829-t002:** In the randomization column, 1: EAW + ES + RT; 0: EAW + delayed-ES + noRT. In the order column, C: control.

Subject ID	Randomization	Assignment	Order in the Group
0881	1	EAW + ES + RT	1
0882	1	EAW + ES + RT	2
0883	0	EAW + delayed-ES + noRT	1C
0884	1	EAW + ES + RT	3
0885	0	EAW + delayed-ES + noRT	2C
0886	1	EAW + ES + RT	4
0887	1	EAW + ES + RT	5
0888	0	EAW + delayed-ES + noRT	3C
0889	0	EAW + delayed-ES + noRT	4C
08810	0	EAW + delayed-ES + noRT	5C
08811	0	EAW + delayed-ES + noRT	6C
08812	0	EAW + delayed-ES + noRT	7C
08813	0	EAW + delayed-ES + noRT	8C
08814	0	EAW + delayed-ES + noRT	9C
08815	1	EAW + ES + RT	6
08816	1	EAW + ES + RT	7
08817	1	EAW + ES + RT	8
08818	1	EAW + ES + RT	9
08819	1	EAW + ES + RT	10
08820	0	EAW + delayed-ES + noRT	10C

**Table 3 jcm-14-01829-t003:** The progression of the closed-chain resistance training using implanted epidural stimulation.

Progression		
Both feet flat on the floor with the knees and hips flexed > 90-degrees	Both feet flat on the floor with the knees and hips flexed = to 90 degrees	Both feet flat on the floor with the knees and hips flexed < 90 degrees
4 sets of 10 reps	4 sets of 10 reps	4 sets of 10 reps

**Table 4 jcm-14-01829-t004:** The number of persons who have enrolled, completed, or withdrawn from the study.

Subject ID	TT + LPWS/NMES	Baseline 1	Post-Intervention	Post-Intervention 2	Sex	LOI	TSI (yrs.)	AIS	Classification
0881	EAW + ES + RT	C	C	C	M	C8	5	A	Tetraplegia
0882	EAW + ES + RT	C	C	X	M	T11	7	B	Paraplegia
0883	EAW + delayed-ES + noRT	C	C	C	M	T6	12	A	Paraplegia
0884	EAW + ES + RT	C	C	C	M	T4	24	A	Paraplegia
0885	Withdrew	X	X	X	M	T7	2	B	Paraplegia
0886	Withdrew	X	X	X	M	T9	3	A	Paraplegia
0887	Screen failed	X	X	X	M	T3	2.5	A	Paraplegia

C = completed the study; X = withdrawn or screen-failed from the study.

## Data Availability

The full study protocol summarized in this paper will be shared by the lead contact author (Ashraf Gorgey (ashraf.gorgey@va.gov)) upon request and after obtaining necessary approval from the local research office.

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
