# Peer review of "Epidural Stimulation and Resistance Training (REST-SCI) for Overground Locomotion After Spinal Cord Injury: Randomized Clinical Trial Protocol"

_jcm, 2025, doi:10.3390/jcm14061829_

Round 1
Reviewer 1 Report
Comments and Suggestions for Authors
Thank you for inviting me in reviewing this manuscript. In this protocol, the authors reported a randomized clinical trial on the effectiveness of resistance training (RT). The topic is of interest, and the method is generally resonable. However, there are still some issues that need to be solved.
1. In the abstract, the authors briefly introduced the background and the rationality of the study. I suggest the authors further clarifying the primary and secondary outcomes of this study.
2. The introduction section is currently too long. If the author focuses on the treatment method of RT, it should revolve around this therapy rather than epidural stimulation.
3. It seems that the authors would like to compare if EAW+ES+RT is better than that without RT. However, in terms of experimental design, the author conducted a relatively complex design, and the control of variables may not be rigorous enough.
4. The images in the article are not clear enough, and some elements may require confirmation of copyright issues. Meanwhile, it is recommended to use elements that are more in line with academic standards.
Author Response
Open Review
( ) I would not like to sign my review report
(x) I would like to sign my review report
Quality of English Language
(x) The quality of English does not limit my understanding of the research.
( ) The English could be improved to more clearly express the research.
|
Yes |
Can be improved |
Must be improved |
Not applicable |
|
|
Does the introduction provide sufficient background and include all relevant references? |
( ) |
(x) |
( ) |
( ) |
|
Is the research design appropriate? |
( ) |
(x) |
( ) |
( ) |
|
Are the methods adequately described? |
(x) |
( ) |
( ) |
( ) |
|
Are the results clearly presented? |
( ) |
(x) |
( ) |
( ) |
|
Are the conclusions supported by the results? |
( ) |
(x) |
( ) |
( ) |
Comments and Suggestions for Authors
Thank you for inviting me in reviewing this manuscript. In this protocol, the authors reported a randomized clinical trial on the effectiveness of resistance training (RT). The topic is of interest, and the method is generally reasonable. However, there are still some issues that need to be solved.
Authors: We would like to thank the reviewer for his time and effort reviewing our manuscript and providing constructive feedback. We addressed your concerns in the order they appear in your letter.
- In the abstract, the authors briefly introduced the background and the rationality of the study. I suggest the authors further clarifying the primary and secondary outcomes of this study.
Authors: Thank you, this was clarified based on your request.
- The introduction section is currently too long. If the author focuses on the treatment method of RT, it should revolve around this therapy rather than epidural stimulation.
Authors: The primary focus is epidural stimulation. Resistance training was added to augment the actions of epidural stimulation on the primary and secondary outcome variables. We totally agree with the reviewers and we have shortened the introduction and also we clarified several points relevant to the design of the triatal.
- It seems that the authors would like to compare if EAW+ES+RT is better than that without RT. However, in terms of experimental design, the author conducted a relatively complex design, and the control of variables may not be rigorous enough.
Authors: The study has two phases. The first 6 months is EAW+ES vs. EAW+no ES. So we are comparing the effect of adding ES to EAW. The second 6 months is EAW+ES+RT compared to EAW+delayed ES+no RT, we are comparing whether the addition of RT to the EAW+ES induces further motor gains compared only to EAW+delayed ES group. We referred to them as delayed ES because, they did not receive ES unit 6 months from the beginning of the study. We agree this is relatively complex design and we believe that each phase was well balanced with an adequate control group.
- The images in the article are not clear enough, and some elements may require confirmation of copyright issues. Meanwhile, it is recommended to use elements that are more in line with academic standards.
Authors: We totally agree with the reviewer that the figures were not at the scientific standard deemed for publication. We have worked to improve the quality of the submitted figures in the current version of the manuscript. We do apologize for any inconvenience.
Reviewer 2 Report
Comments and Suggestions for Authors
Reviewer’s comments:
Summary:
The study protocol, “Epidural Stimulation and Resistance Training (REST-SCI) for Overground Locomotion after Spinal Cord Injury. A Protocol 3 of Randomized Clinical Trial”, details the experimental design for a clinical study that examines the combinatory impact of exoskeleton-assisted walking, percutaneous epidural stimulation, and resistance training. In addition to measures of locomotor function including electromyography, overground walking, and walking index for spinal cord injury scale, the authors also propose to examine cardiovascular function, urinary function, body composition, and blood biomarkers to determine if the addition of resistance training positively impacts long-term outcomes for persons with ASI-A and B spinal cord injury. Overall, this protocol is comprehensive in its measures of experimental outcomes, and the authors should be commended for their experimental design. Despite the promise that this protocol has there is room for significant improvement in the presentation of the paper’s methods, and this publication would be more impactful if ideas were presented more consistently and if the quality of figures was improved.
Article:
The general study design is thorough and does an excellent job of touching on the many physiological changes that can occur following the addition of RT to rehabilitation protocols involving ES for SCI. Of note, however, is the absence of autonomic dysreflexia (AD) in the introduction and missing information about how AD will be treated if an event occurs during training, or urinary function studies. This reviewer highly suggests that AD be included in the discussion of cardiovascular deficits seen in persons with SCI and in the tests used to examine orthostatic hypotension. Greater clarity on the status of this study early on (in the introduction) will also improve the reader’s overall understanding of the nature of this publication. This publication has promise to be highly impactful, provided that the authors do some careful work during revisions.
Specific comments:
1. The figure quality is poor and most figures will need to be provided at a higher resolution so that they are legible. Overall, it may also be a good idea to combine several of the figures to shorten the paper and make it more digestible for readers.
2. The text should be carefully proofread to improve the consistency of the tense in many of the sections. This confusion may be because the study is ongoing, and some aspects of the study have been performed while others have not. Please work on the consistency here so that the status of the work is more apparent.
3. The methods section could be improved with greater consistency in subheadings and numbering. Three different types of numbering/bullet points are used throughout the paper, and this can make the study challenging to follow. I suggest the authors use bolded subheadings followed by numbers below subheadings. Of note is the section on urinary function labeled as “1” while the heading directly below it returns to letters “m.”
4. Table 1 would benefit from the addition of horizontal lines as it is difficult to determine which outcome measures are attributed to which study.
5. Line 145: This first mention of NMES needs to be defined.
6. Line 332: “If the ES is off, subjects with training…” This sentence does not seem like it’s placed appropriately in the methods as it refers to outcomes. Perhaps it should be moved to the discussion or removed.
7. Line 438, Section C: it is unclear whether the tests for overground ambulation are conducted with or without ES.
8. In the methods sections discussing BMI, DEXA scans, BMR tests, blood tests and NIRS, the timing of tests would be helpful to include. Although it may be stated elsewhere in the paper, the frequency of tests would improve clarity.
9. The section on urodynamics is the first mention of alternate ES configurations. The introduction gives the impression that ESA + ES (for locomotion) + RT could improve urinary function outside of the use of ES for the lower urinary tract. It is important to clarify whether you are looking for improvements in urinary function due to the training paradigm alone or due to the training paradigm plus the addition of bladder-specific ES. Either would be relevant as locomotor training alone has been found to improve urinary function, however this clarification is important. If alternate ES paradigms are used, please specify the stimulation frequency, location, and intensity
10. If alternate ES configurations were used, is the paradigm changed when measuring orthostatic hypotension during tilt-table tests?
11. Line 665: This section on tilt-table outcomes is in the wrong section, rather than being in the statistical analysis section it should be grouped with the tilt-table methods.
12. The statistical analysis section appears to only comment on tilt-table analysis and needs to be expanded or altered to be more comprehensive.
Author Response
Open Review
(x) I would not like to sign my review report
( ) I would like to sign my review report
Quality of English Language
(x) The quality of English does not limit my understanding of the research.
( ) The English could be improved to more clearly express the research.
|
Yes |
Can be improved |
Must be improved |
Not applicable |
|
|
Does the introduction provide sufficient background and include all relevant references? |
(x) |
( ) |
( ) |
( ) |
|
Is the research design appropriate? |
(x) |
( ) |
( ) |
( ) |
|
Are the methods adequately described? |
( ) |
(x) |
( ) |
( ) |
|
Are the results clearly presented? |
( ) |
( ) |
( ) |
(x) |
|
Are the conclusions supported by the results? |
( ) |
( ) |
( ) |
(x) |
Comments and Suggestions for Authors
Reviewer’s comments:
Summary:
The study protocol, “Epidural Stimulation and Resistance Training (REST-SCI) for Overground Locomotion after Spinal Cord Injury. A Protocol 3 of Randomized Clinical Trial”, details the experimental design for a clinical study that examines the combinatory impact of exoskeleton-assisted walking, percutaneous epidural stimulation, and resistance training. In addition to measures of locomotor function including electromyography, overground walking, and walking index for spinal cord injury scale, the authors also propose to examine cardiovascular function, urinary function, body composition, and blood biomarkers to determine if the addition of resistance training positively impacts long-term outcomes for persons with ASI-A and B spinal cord injury. Overall, this protocol is comprehensive in its measures of experimental outcomes, and the authors should be commended for their experimental design. Despite the promise that this protocol has there is room for significant improvement in the presentation of the paper’s methods, and this publication would be more impactful if ideas were presented more consistently and if the quality of figures was improved.
Authors: We would like to thank the reviewer for his time and effort to provide constructive feedback regarding our study protocol. We have attempted to follow your recommendation to ensure improvement in the presentation of the work.
Article:
The general study design is thorough and does an excellent job of touching on the many physiological changes that can occur following the addition of RT to rehabilitation protocols involving ES for SCI. Of note, however, is the absence of autonomic dysreflexia (AD) in the introduction and missing information about how AD will be treated if an event occurs during training, or urinary function studies.
This reviewer highly suggests that AD be included in the discussion of cardiovascular deficits seen in persons with SCI and in the tests used to examine orthostatic hypotension. Greater clarity on the status of this study early on (in the introduction) will also improve the reader’s overall understanding of the nature of this publication. This publication has promise to be highly impactful, provided that the authors do some careful work during revisions.
Authors: We have included an entire section about AD in the discussion section highlighting the current knowledge about applications of epidural stimulation and AD. We have also modified the introduction based on your recommendations.
Specific comments:
- The figure quality is poor and most figures will need to be provided at a higher resolution so that they are legible. Overall, it may also be a good idea to combine several of the figures to shorten the paper and make it more digestible for readers.
We totally agree with the reviewer that the figures were not at the scientific standard deemed for publication. We have worked to improve the quality of the submitted figures in the current version of the manuscript. We do apologize for any inconvenience.
- The text should be carefully proofread to improve the consistency of the tense in many of the sections. This confusion may be because the study is ongoing, and some aspects of the study have been performed while others have not. Please work on the consistency here so that the status of the work is more apparent.
Authors: We have proofread our manuscript to ensure consistency of the tense across the entire text, since the study is still actively ongoing.
- The methods section could be improved with greater consistency in subheadings and numbering. Three different types of numbering/bullet points are used throughout the paper, and this can make the study challenging to follow. I suggest the authors use bolded subheadings followed by numbers below subheadings. Of note is the section on urinary function labeled as “1” while the heading directly below it returns to letters “m.”
Authors: We totally agree with the reviewer and the method section into three major sections
- Intervention
- Training
- Measurements
- Recruitment strategies
- Statistical analyses
- Table 1 would benefit from the addition of horizontal lines as it is difficult to determine which outcome measures are attributed to which study.
Authors: Thank you, we have separated different studies to ensure that the outcomes represent the corresponding studies.
- Line 145: This first mention of NMES needs to be defined.
Authors: Thank you, this was defined.
- Line 332: “If the ES is off, subjects with training…” This sentence does not seem like it’s placed appropriately in the methods as it refers to outcomes. Perhaps it should be moved to the discussion or removed.
Authors: Thank you, the sentence was totally deleted based on your recommendation.
- Line 438, Section C: it is unclear whether the tests for overground ambulation are conducted with or without ES.
Authors: Overground ambulation is achieved by one person and was conducted with SCES on. We cannot conduct it with SCES off
- In the methods sections discussing BMI, DEXA scans, BMR tests, blood tests and NIRS, the timing of tests would be helpful to include. Although it may be stated elsewhere in the paper, the frequency of tests would improve clarity.
Authors: Thank you, all these measurements were conducted at baseline, post-intervention 1 and post-intervention 2. These measurements were 6 months apart.
- The section on urodynamics is the first mention of alternate ES configurations. The introduction gives the impression that ESA + ES (for locomotion) + RT could improve urinary function outside of the use of ES for the lower urinary tract. It is important to clarify whether you are looking for improvements in urinary function due to the training paradigm alone or due to the training paradigm plus the addition of bladder-specific ES.
Authors: We are looking for improvement in bladder function as result of the training paradigm. We do not conduct additional bladder specific ES.
Either would be relevant as locomotor training alone has been found to improve urinary function, however this clarification is important. If alternate ES paradigms are used, please specify the stimulation frequency, location, and intensity.
Authors: We do not use additional bladder training. We have clarified this point under the urodynamic section.
- If alternate ES configurations were used, is the paradigm changed when measuring orthostatic hypotension during tilt-table tests?
Authors: This has been clarified. The participant is initially in supine lying for 5 min followed by an attempt to perform a head-up tilt using a motorized tilting table for 10 min. The tilting maneuver was discontinued if systolic blood pressure (SBP) dropped by 40 mm Hg from resting baseline. This is followed by repeating the same protocol with SCES on after a 30-min washout period. The SCES was turned on 3–5 min in supine lying before performing the head-up tilt maneuver.
- Line 665: This section on tilt-table outcomes is in the wrong section, rather than being in the statistical analysis section it should be grouped with the tilt-table methods.
Authors: We totally agree, and this has moved to the tilt table section.
- The statistical analysis section appears to only comment on tilt-table analysis and needs to be expanded or altered to be more comprehensive.
Authors: We totally agree with the reviewer, and we have fixed this concern in the revised version.
Round 2
Reviewer 2 Report
Comments and Suggestions for Authors
Thank you for taking the time to carefully consider my comments. This paper is greatly improved.
Comments on the Quality of English LanguageThe manuscript is clearly written and understandable.